# Analyzing the Antinociceptive Effect of Interleukin-31 in Mice

**DOI:** 10.3390/ijms241411563

**Published:** 2023-07-17

**Authors:** Iwao Arai, Minoru Tsuji, Kohei Takahashi, Saburo Saito, Hiroshi Takeda

**Affiliations:** 1Department of Pharmacology, International University of Health and Welfare, 2600-1 Kitakanemaru, Ohtawara 324-8510, Japantakahashi-k@iuhw.ac.jp (K.T.);; 2Division of Environmental Allergy, The Jikei University School of Medicine, 3-25-8 Nishi-Shinbashi, Tokyo 105-8461, Japan; misaburo@jikei.ac.jp

**Keywords:** analgesia, alloknesis, antinociception, interleukin-31 (IL-31), interleukin receptor A (IL-31RA), IL-31 receptor A-deficient (IL-31RAKI) mice, itch, pain

## Abstract

The theory that an itch inhibits pain has been refuted; however, previous research did not investigate this theory for an interleukin-31 (IL-31)-induced itch. Previously, we have found that morphine-induced antinociception was partially reduced in IL-31 receptor A (IL-31RA)-deficient (IL-31RAKI) mice, indicating that IL-31RA may play an important role in morphine-induced peripheral antinociception. In the present study, we evaluated the effect of IL-31-induced analgesia on a 2,4,6-trinitrochlorobenzene (TNCB)-sensitized mice using a hot-plate test. This test evaluated the antinociceptive activity of morphine and non-steroidal anti-inflammatory drugs (NSAIDs). Repeated pretreatment with IL-31 showed significant antinociceptive action. Furthermore, its combination with morphine, but not with NSAIDs, increased the analgesic action. In contrast, treatment with TNCB and capsaicin decreased antinociception. Moreover, TNCB increased IL-31RA expression in the dorsal root ganglia at 24 h, whereas capsaicin inhibited it. The comparative action of several analgesics on TNCB or capsaicin was evaluated using a hot-plate test, which revealed that the antinociceptive activity was decreased or disappeared in response to capsaicin-induced pain in IL-31RAKI mice. These results indicate that the analgesic action of IL-31 involves the peripheral nervous system, which affects sensory nerves. These results provide a basis for developing novel analgesics using this mechanism.

## 1. Introduction

Interleukin-31 (IL-31), a possible mediator of itching, induces severe pruritus and dermatitis in mice [1]. Elevated cutaneous IL-31 expression levels have been observed in lesional skin atopic dermatitis [2,3]. Moreover, the repeated administration of IL-31 causes itch-associated scratching behavior, which is significantly increased with the increased expression of IL-31 receptor A (IL-31RA) in the dorsal root ganglia (DRG) [4]. The interaction between cutaneous IL-31 and neuronal DRG IL-31RA causes severe itch-associated scratching behavior (long-lasting scratching, LLS) [5].

The central and peripheral nervous systems (CNS and PNS, respectively) are well-known sites of action for analgesia. Morphine, an opioid, induces analgesia by acting on both the CNS and PNS [6]. It inhibits the release of neurotransmitters from the primary afferent terminals in the spinal cord and activates the descending inhibitory controls in the midbrain [7,8]. However, the site of action of morphine in the PNS and its role in regulating pain transmission remain unclear. The administration of morphine increases the scratch counts within 10 min, followed by a return to basal levels approximately 90 min after administration. A close correlation has been observed between time-course changes in morphine-induced LLS counts and antinociceptive activity [9]. Moreover, IL-31 causes characteristic LLS upon administration [5]. In IL-31RA-deficient (IL-31 receptor A^tLacZ/+^ knock-in) mice, the administration of morphine results in the disappearance of LLS and the partial disappearance of antinociceptive activity [9]. Additionally, recent findings have indicated that IL-31 is partially involved in the peripheral analgesic mechanism. As IL-31 and IL-31RA are not expressed in the CNS, the endogenous opioid system is activated under pathological conditions. Moreover, morphine-induced antinociceptive activity in the peripheral site is better than that in the central site of morphine action because central untoward adverse effects, such as respiratory depression, somnolence, and addiction, are avoided [10,11]. Although non-steroidal anti-inflammatory drugs (NSAIDs) also mediate a peripheral analgesic effect, they inhibit peripheral inflammation. In contrast, morphine has no anti-inflammatory action on the peripheral site. 

The hot-plate test using mice is simple and easy to perform [12] and is generally used for strong-acting analgesics, such as opioids, but not for peripherally acting drugs. Currently, a few experimental models can evaluate peripherally acting analgesic drugs in mice. A well-known model for analgesic action for NSAIDs is paw inflammation induced by an injection of Freund’s complete adjuvant [13] or carrageenin [14] and a subsequent hyperalgesia assessment in rats [15,16]. However, it is substantially more difficult to perform long-term tests in mice. Moreover, although these methods cause inflammation, they can be used to evaluate even weak analgesics, such as NSAIDs and acetaminophen, if the pain threshold is lowered. Acute contact dermatitis is induced after a single application of hapten, dinitrochrolbenzen (DNCB) [17]. A modified hot-plate test on the DNCB derivative 2,4,6-trinitrochlorobenzene (TNCB) has been previously performed [18]. TNCB was applied to the limbs of mice to reduce the pain threshold, which allowed the evaluation of antinociceptive activity at a temperature lower than that used in the hot-plate test [19]. Additionally, it was possible to evaluate the antinociceptive activity of NSAIDs and IL-31 using TNCB. 

Itching elicits a strong desire to scratch; therefore, scratching behavior count is a useful index to evaluate itching [20]. Mice exhibit two types of scratching behavior, long-lasting scratching (LLS, scratching behavior lasting more than 1.0 s) and short-lasting scratching (SLS, scratching behavior lasting from 0.3 to 1.0 s). Current studies on itching are based on the human perceptive sense, and these nociceptive stimuli have no discernible differences. However, the sensory perception of a foreign substance and true itching could be differentiated by dividing the scratching behavior of mice into LLS and SLS [21]. The number of spontaneous scratches can be automatically detected and objectively evaluated via a computer. Based on our evaluation standard, we have previously found that histamine is not a pruritogen [22].

Capsaicin acts on the transient receptor potential cation channel vanilloid subfamily V member 1 (TRPV1), which modulates nociceptive inputs to the spinal cord and brain stem centers integrating diverse painful stimuli [23,24]. The sensation of itching can be reduced by the painful sensations caused by scratching [25]. The inhibition of itching through painful stimuli has been experimentally demonstrated using various painful stimuli. NC/Nga mice, an animal model of atopic dermatitis with chronic itching, spontaneously develop skin lesions [26]. We have previously demonstrated that cutaneous prostaglandins (PGs) levels are significantly elevated upon scratching the mouse skin with a stainless-steel wire brush (mechanical scratching), and these PGs suppressed LLS in skin-lesioned NC/Nga mice [27,28]. Because PGs are associated with inflammation, their administration enhances pain [29]. Notably, although the pain-induced suppression of itching is temporary, the effect of capsaicin lasts more than 72 h, suggesting that this effect may not only be due to pain but also another action of capsaicin. The partial activation of TRPV1 by capsaicin results in pain, which, in turn, may partially suppress itching sensations [30]. Paradoxically, the application of capsaicin produces transient burning pain and induces analgesia for neuropathic pain. The mechanisms underlying these opposing actions of capsaicin on the onset of pain and the induction of analgesia have not yet been explained. The application of capsaicin causes the desensitization of TRPV1 [31], the inhibition of nociceptor firing [32], and a decrease in mechanotransduction [33]. These early effects of capsaicin on the function of primary afferents might contribute to the analgesic effects immediately after capsaicin application. However, these effects may be unrelated to suppressing itching sensations because there are no reports of an analgesic action inhibiting itching for more than 72 h following capsaicin application. Recently, we reported that capsaicin suppresses LLS and SLS for more than 72 h; at this point, the expression of DRG IL-31RA mRNA is significantly decreased, whereas that of cutaneous IL-31RA shows no significant change. Thus, capsaicin may suppress LLS by inhibiting IL-31RA mRNA expression in the DRG [34].

Our previous report suggested the partial involvement of IL-31 in the antinociceptive action of morphine [9]. Moreover, we investigated the effects of pretreatment with IL-31 on morphine-induced antinociceptive activity using a modified hot-plate test and found a close correlation between morphine-induced LLS and antinociceptive effect. We also investigated LLS as an indicator of itching by morphine-induced itching and found that the repeated administration of IL-31 gradually promoted LLS in a dose-dependent manner and increased DRG neuronal IL-31RA expression. These data show that cutaneously-injected IL-31 induces LLS and promotes DRG IL-31RA expression [4]. 

The present study aimed to reveal the mechanism of action of IL-31 on TNCB- or capsaicin-induced pain models and assess the long-term physiological effects of its application. Our results provide important insights into the development of a new type of algesic based on the antinociceptive mechanism of IL-31.

## 2. Results

### 2.1. Effect of IL-31 on TNCB Applied Hot-Plate Test in BALB/c Mice

In the conventional hot-plate test (51 °C), IL-31 did not show significant antinociceptive activity. In contrast, in the TNCB-applied (3%, 45 °C) hot-plate test [19], the latency rapidly decreased 30 min after the TNCB application (Figure 1a, red line). Moreover, a single pretreatment injection with IL-31 (50 μg/kg, intraperitoneal) showed slight antinociception, which was not significant compared with that of the vehicle (phosphate-buffered saline, PBS, 10 mL/kg)-treated group. However, repeated pretreatment with IL-31 (50 μg/kg, intraperitoneal, every 12 h for 3 days) significantly increased the antinociceptive action on the TNCB-applied hot-plate test at 45 °C (Figure 1a, green line) and the total antinociceptive index of IL-31 after 0.5–6 h (AUC_0.5–6 h_) of TNCB application compared with the vehicle (acetone–ethanol mixed liquor, AEM)-treated group (Figure 1b, red column). These results suggest that the repeated administration of IL-31 was a suitable condition for the evaluation of the antinociceptive effect. Therefore, we performed subsequent experiments with repeated administration of IL-31 (50 μg/kg; Figure 1a, intraperitoneal, every 12 h for 3 days).

In the vehicle (PBS, 10 mL/kg, intraperitoneal) + AEM-treated group, a few LLS counts were observed (Figure 1c, blue line). And, these LLS counts decreased after TNCB, which was not significant compared with those of the AEM-treated group (Figure 1c, yellow line). On the other hand, a single large dose injection of IL-31 (1 mg/kg, intraperitoneal) increased LLS counts, which gradually increased 4 h after IL-31 administration. This increase in LLS counts showed a circadian rhythm; in particular, the LLS counts significantly increased at nighttime. Moreover, the application of 0.2 mL of TNCB to the dorsal skin surface significantly decreased LLS counts immediately after or within 6 h after application (Figure 1c, red line). The total LLS counts also significantly decreased in the IL-31 + TNCB group compared with those in the IL-31 + AEM-treated group (Figure 1d, yellow column). However, the total LLS counts of IL-31 + TNCB were not significantly changed compared with those in the PBS + TNCB-treated group.

### 2.2. Effects of the Combination of Morphine and IL-31 on Hot-Plate Test in BALB/c Mice

In a conventional hot-plate test (51 °C), morphine (MP, 3 mg/kg, subcutaneously) significantly increased the latency (Figure 2a, green line) and total antinociceptive index 15–120 min after administration (AUC_15–120 min_) (Figure 2b, green column) compared with the saline-treated group (Figure 2b, blue column). Morphine (3 mg/kg, subcutaneously) significantly increased the latency (Figure 2a, green line) and total antinociceptive index 15–120 min after administration (AUC_15–120 min_) (Figure 2b, green column) compared with the saline-treated group (Figure 2b, blue column) in a conventional hot-plate test (51 °C). In contrast, repeated pretreatment with IL-31 decreased latency (Figure 2a, red line) and the total antinociceptive index after 15 to 120 min (AUC_15–120 min_) (Figure 2b, red column); however, no significant difference was observed. Moreover, the effect of the combination of morphine and repeated pretreatment with IL-31 significantly enhanced morphine-induced increasing latency (Figure 2a, yellow line) and the total antinociceptive index (AUC_15–120 min_) (Figure 2b, yellow column).

### 2.3. Effects of the Combination of IL-31 and Loxoprofen on TNCB-Applied Hot-Plate Test in BALB/c Mice

In the group where the vehicle (acetone–ethanol mixed liquor, AEM, 0.04 mL/each limb) was applied to the limbs of the mice, latency was not changed during the experimental period on the modified hot-plate test (45 °C) (Figure 3a, black line). In the TNCB (1%)-applied hot-plate (45 °C) test, the latency significantly decreased from 30 to 120 min after TNCB application (Figure 3a, blue line). Loxoprofen (LXP, 15 mg/kg, oral) significantly increased antinociception (Figure 3a, green line) and the total antinociceptive index after 30–120 min (AUC_30–120 min_) (Figure 3b, green column) compared with the vehicle (carboxymethyl cellulose sodium, CMC, oral)-treated group (Figure 3a, blue line: Figure 3b, blue column). Repeated pretreatment with IL-31 also increased latency (Figure 3a, red line) and total antinociceptive index (AUC_30–120 min_) (Figure 3b, red column). However, the combination of loxoprofen and IL-31 did not significantly enhance latency (AUC_30–120 min_) (Figure 3a, yellow line) and total antinociceptive index (AUC_30–120 min_) (Figure 3b, yellow column) compared with treatment with loxoprofen alone.

In the preliminary study, we assessed the time-course change of cutaneous temperature as an indicator of fever, and skin weight as an indicator of swelling, after the 1% TNCB application to determine the optimal experimental conditions. The cutaneous temperature decreased within 1 to 6 h after TNCB application, then significantly increased after 24 h of application and returned to basal level after 72 h. Skin tissue weight significantly increased between 6 and 72 after TNCB application. 

Therefore, we measured the cutaneous weight and temperature for 24 h after TNCB application in subsequent experiments. Then, IL-31 did not change the cutaneous temperature or swelling. In contrast, loxoprofen (LXP) significantly decreased TNCB-induced fever and swelling (Figure 3c,d).

### 2.4. Comparison of the Effects of TNCB or Capsaicin on Modified Hot-Plate Test in BALB/c Mice

The application of TNCB (3%, 0.04 mL/site of limb) or capsaicin (1%, 0.04/site of limb) rapidly decreased latency on the hot-plate (45 °C) test within 0.5 to 6 h after their topical application (Figure 4a, red and green lines). Cutaneous prostaglandin D_2_ (PGD_2_) contents significantly increased 1 h after TNCB application, whereas this effect was not observed in the capsaicin-applied group (Figure 4b). Capsaicin significantly suppressed IL-31RA expression in the DRG but not the TNCB applications groups (Figure 4c).

TNCB markedly increased cutaneous IL-31 mRNA expression 1 h after TNCB application; however, the difference was not significant (Figure 4d). Moreover, capsaicin significantly decreased TRPV1 expression in the DRG but not the TNCB application group at 6 h after these applications (Figure 4e). The TNCB and capsaicin groups exhibited pain and had shortened latency from 0.5 to 6 h after application, similarly to the effects observed at 6 h; however, many differences were observed in the expression of several pain-related factors. Therefore, we investigated the site of action of IL-31 on PGD_2_- or capsaicin-induced pain in mice.

### 2.5. Effect of IL-31 on ProstaglandinD_2_- or Capsaicin-Applied Modified Hot-Plate Test in BALB/c Mice

In the group where the vehicle (ethanol, 0.04 mL/each limb) was applied to the limbs of the mice, latency was not changed during the experimental period on the modified hot-plate (45 °C) test (Figure 5a,c, blue line). Similar to the group receiving repeated pretreatment with IL-31, latency was also not changed during the experimental period (Figure 5a,c, red line). In contrast, prostaglandin D_2_ (PGD_2_) significantly decreased latency from 0.5 to 4 h after PGD_2_ application (Figure 5a, green line). After repeated pretreatment with IL-31 in the PGD_2_-applied group, the PGD_2_-induced decreased latency was significantly increased from 0.5 to 1 h after PGD_2_ application (Figure 5a, yellow line). The total antinociceptive index of IL-31 after 30 to 120 min (AUC_30–120 min_) after vehicle (PBS) application was not significantly different between the PBS and IL-31 pretreatment groups (Figure 5b, left side). In contrast, the total antinociceptive index of IL-31 after 30 to 120 min (AUC_30–120 min_) in the PGD_2_-applied group showed a significant difference between the PBS and IL-31 pretreatment groups (Figure 5b, right side).

Capsaicin significantly decreased latency from 0.5 to 6 h after application (Figure 5c, green line). After repeated pretreatment with IL-31 in the capsaicin-applied group, the capsaicin-induced decreased latency did not change (Figure 5c, yellow line). The total antinociceptive index of the vehicle (PBS)-applied to the group and the IL-31 pretreatment group were not significantly different (Figure 5d, left side). Therefore, we investigated the effects of several analgesics on TNCB- or capsaicin-induced pain in wild-type and IL-31RAKI mice.

### 2.6. Effects of Morphine on TNCB- or Capsaicin-Applied Hot-Plate Test in Wild-Type and IL-31RAKI Mice

In the TNCB-applied hot-plate test (45 °C), the latency rapidly decreased 30 min after TNCB (3%, 0.04 mL/each limb) and capsaicin (1%, 0.04 mL/each limb) application (Figure 6a, blue and green lines) in BALB/c mice. Morphine (3 mg/kg, subcutaneously) delayed the latency and the increased antinociceptive index 30 to 120 min after administration (AUC_30–120 min_) compared with those of the vehicle (saline)-treated group (Figure 6a, red and yellow lines). The total antinociceptive index of morphine after 30 to 120 min (AUC_30–120 min_) was approximately the same as in those of the TNCB- and capsaicin-applied hot-plate test groups. However, there was a significant effect for the TNCB-applied hot-plate test (Figure 6b, left side), but not the capsaicin-applied hot-plate test (Figure 6b, right side). 

In the conventional hot-plate test (51 °C), the latency did not change during the experimental period (Figure 6c, blue and green lines) in wild-type (C57BL/6) and IL-31RAKI mice (C57BL/6 genetic background). Morphine (3 mg/kg, subcutaneously) delayed the latency and increased antinociception compared with the vehicle-treated group (Figure 6c, red and yellow lines). The total antinociceptive index of morphine after 30 to 120 min (AUC_30–120 min_) was decreased in IL-31RAKI mice compared with wild-type mice in the conventional hot-plate test. In contrast, a significant effect was not observed in IL-31RAKI mice (Figure 6d, right side).

### 2.7. Effects of Loxoprofen on TNCB- or Capsaicin-Applied Hot-Plate Test in Wild-Type and IL-31RAKI Mice

The latency rapidly decreased 30 min after TNCB or capsaicin was applied in the hot-plate (45 °C) test in BALB/c mice (Figure 7a, blue and green line). Loxoprofen (LXP), which is a reversible cyclooxygenase inhibitor, (15 mg/kg, oral), significantly delayed latency 0.5 to 6 h after loxoprofen treatment on the TNCB (3%, 0.04 mL/each limb)-applied hot-plate test, compared with the respective value of the vehicle (carboxymethyl cellulose sodium, CMC, 10 mL/kg, oral)-treated group (Figure 7a, red lines).

However, the total antinociceptive index of loxoprofen after 0.5 to 6 h (AUC_0.5–6 h_) was not significant increased compared with that of the CMC-treated group (Figure 7b, left side). However, loxoprofen did not affect the capsaicin-applied hot-plate test compared with the CMC-treated group (Figure 7a,b, red line and column). On the other hand, loxoprofen did not affect the capsaicin-applied hot-plate test (Figure 7a,b, yellow line and column, right side).

In wild-type (C57BL/6) and IL-31RAKI (C57BL/6 genetic background) mice, the latency time rapidly decreased 30 min after TNCB application (Figure 7c, blue and green lines). Loxoprofen (15 mg/kg, oral) significantly delayed the latency at 2 h after treatment and increased antinociception compared with the respective value of the CMC-treated group (Figure 7c, red line). The total antinociceptive index of loxoprofen after 0.5 to 6 h (AUC_0.5–6 h_) was significantly increased compared with that of the CMC-treated group (Figure 7d, left side). However, loxoprofen did not affect the TNCB-treated hot-plate test in IL-31RAKI mice (Figure 7c,d, yellow line and column, right side). These results were similar to those of capsaicin-induced pain on the modified hot-plate test (Figure 7b).

### 2.8. Effects of Acetaminophen on TNCB- or Capsaicin-Applied Hot-Plate Test in Wild-Type and IL-31RAKI Mice

The latency rapidly decreased 30 min after TNCB- or capsaicin-applied hot-plate (35 °C) tests in BALB/c mice (Figure 8a, blue and green lines). Acetaminophen (APAP, 300 mg/kg, oral) delayed the latency and increased antinociceptive activity on the TNCB-applied hot-plate test, compared with the vehicle (carboxymethyl cellulose sodium, CMC, 10 mL/kg, oral)-treated group (Figure 8a, red lines). The total antinociceptive index of acetaminophen after 30 to 120 min (AUC_30–120 min_) was significantly increased compared with that of the CMC-treated group (Figure 8b, red line and column, left side). However, acetaminophen did not affect the capsaicin-treated hot-plate test, compared with the CMC-treated group (Figure 8a,b, yellow line and column, right side).

In wild-type (C57BL/6) and IL-31RAKI (C57BL/6 genetic background) mice, the latency rapidly decreased 30 min after TNCB application on the modified hot-plate (35 °C) test (Figure 8c, blue and green lines). Acetaminophen (APAP, 300 mg/kg, oral) delayed the latency time and increased antinociception compared with the CMC-treated group (Figure 8c, red line). The total antinociceptive index of acetaminophen after 30–120 min (AUC_30–120 min_) was significantly increased compared with that of the CMC-treated group (Figure 8d, left side). However, acetaminophen did not affect the TNCB-applied hot-plate test in IL-31RAKI mice (Figure 8d, yellow line and column, right side). These results were similar to those of capsaicin-induced pain on the modified hot-plate test (Figure 8b).

### 2.9. Long-Term Changes after TNCB or Capsaicin Application on Sense of Itch and Pain 

In the group where the vehicle (acetone-ethanol mixed liquor, AEM, 0.04 mL/each limb) was applied to the limbs of the mice, latency was not changed during the experimental period on the modified hot-plate test (45 °C) (Figure 9a, blue line). Both TNCB (3%, 0.04 mL/each limb) and capsaicin (1%, 0.04/each limb) induced a decreased latency after their application until 6 h. However, their effect differed 24 h after their topical application. The decreased latency was reversed in the TNCB group after 24 h and showed pain torpor until 72 h, whereas that in the capsaicin group continued to decrease until 72 h after application (Figure 9a, red and green lines). In IL-31RAKI mice, TNCB showed decreased latency after application until 6 h (Figure 5a, yellow line) and then returned to basal levels after 24 h (Figure 9a, yellow line). The pain torpor was not observed in IL-31RAKI mice. In the TNCB group, IL-31RA expression in the DRG increased within 1 to 6 days after application, whereas that in the capsaicin group continued decreasing until 6 days after application (Figure 9b, red and green lines). TNCB showed a significant decrease in LLS counts from immediately after application until 6 h (Figure 9c), which then increased 3 days after administration, and gradually decreased to basal levels 12 days after administration (Figure 9c, red line). The capsaicin group also showed significantly decreased LLS counts immediately after its application until 6 days after (Figure 9c, green line), which did not recover during the experimental period. We measured IL-31-induced (1 mg/kg, intravenously) LLS counts for IL-31 reactivity. Before capsaicin application, LLS counts of the vehicle (PBS) and IL-31 administration groups were 70.3 ± 11.3 and 299.1 ± 39.7 counts/24 h, respectively (Figure 9, green and blue columns). The IL-31-induced LLS counts at 1, 24, 72, and 144 h (6 days) after capsaicin application were 30.1 ± 4.3, 56.8 ± 23.1, 184.6 ± 2.2, and 124.2 ± 26.2 counts/24 h, respectively (Figure 9d, red columns), which were significantly decreased compared with those before capsaicin application (Figure 9d, red columns).

## 3. Discussion

The IL-31-caused scratching behavior is distinct from other pruritogens (e.g., histamine, serotonin)-induced scratching [21,35,36,37]. A single intradermal, or other administration route (e.g., intravenously), injection of IL-31 elicits LLS but not SLS, with LLS manifesting gradually approximately 1 h after injection and persisting over 24 h [35]. However, other pruritogens, injected only via an intradermal route, elicits SLS but not LLS, with SLS manifesting immediately after the intradermal injection and persisting for at least 30 min. IL-31 is a neuronal transmitter that causes aloknesis but is not a pruritogens, indicating that it causes alloknesis-induced itching [38,39]. In a previous study, we showed that repeated administration of IL-31 gradually increased the LLS counts. Our results indicated that IL-31-induced LLS counts depend on the interaction of blood IL-31 concentration, DRG neuronal IL-31RA expression induces LLS, and IL-31 promotes the onset of DRG neuronal IL-31RA [5]. In this study, a single pretreatment with IL-31 (50 μg/kg, intrapretoneal) showed no significant antinociceptive activity whereas repeated pretreatment with IL-31 showed significant antinociceptive activity on the TNCB-applied hot-plate test at 45 °C. The repeated administration of IL-31 enhanced the action of IL-31-induced increases in DRG neuronal IL-31RA expression. Moreover, a single large dose of IL-31 (1 mg/kg, intraperitoneal) elicited clear LLS, and TNCB-induced pain inhibited the IL-31-induced LLS. These results suggest that the IL-31-induced itch inhibits TNCB-induced pain. Although it is well-known that pain inhibits itching [40], the reverse phenomenon that an itch inhibits pain is controversial. Our results show that IL-31-induced alloknesis inhibits pain. These data suggest that the sensation of pain and itch may be regulated through their functional antagonism.

In the present study, the repeated pretreatment of IL-31 decreased latency; however, this result was insignificant in the conventional hot-plate test (51 °C). This result suggests that IL-31-induced itching is caused by heat stimulation; therefore, pain and itching actions are indistinguishable. However, morphine-induced antinociception was enhanced with pretreatment with IL-31, suggesting that the increased analgesic action was caused by the IL-31-induced peripheral action of morphine and its interaction with the central analgesic action of morphine. Morphine produces analgesia by acting on both the CNS and PNS, and it is a one of the few drugs to induce LLS counts and cause alloknesis simultaneously [41,42]. In addition, morphine-induced LLS and antinociceptive effects are closely correlated in mice; morphine-induced LLS and antinociceptive effects were completely and partially inhibited in IL-31RAKI mice, respectively [9]. Previously, we reported that IL-31 might play a significant role in the modulation of peripheral morphine-induced antinociception by sensory neurons in IL-31RAKI mice compared with wild-type mice [9]. In the present study, we demonstrated that repeated pretreatment with IL-31 showed antinociceptive activity. These results suggest that the antinociceptive activity of IL-31 partially contributes to the antinociceptive action of morphine.

In the present study, loxoprofen, a PGs inhibitor, significantly increased latency and total antinociceptive activity in the TNCB (1%)-applied hot plate (45 °C) test. The application of 3% TNCB will show almost the same result. Repeated pretreatment with IL-31 also increased latency and total nociception; however, the combined effect of loxoprofen and IL-31 was not significantly enhanced. These results suggest that the antinociceptive activity of IL-31 was approximately as effective as that of loxoprofen (NSAIDs). An enhanced action was not observed because the compound characteristics have the same peripheral action site. Inflammatory reactions increased tissue PGs’ (PGD_2_, PGE_2_, and PGI_2_) biosynthesis and are characterized by redness (arterial extension), fever (increased body temperature), and edema (swelling). Therefore, we examined the comparative effects of IL-31 and loxoprofen on TNCB-induced fever (surface body temperature) and swelling (cutaneous weight) in mice. TNCB significantly increased the cutaneous temperature and skin tissue weight 24 h after application, whereas IL-31 did not change the cutaneous temperature and swelling. In contrast, loxoprofen significantly decreased TNCB-induced fever and swelling. These results show that the main action of IL-31 is not to inhibit PGs biosynthesis. Moreover, even if they share the same peripheral analgesic sites of action, NSAIDs and IL-31 have different mechanisms of action; NSAIDs suppress inflammation by inhibiting the biosynthesis of these PGs. However, these inflammatory reactions are spontaneous defense mechanisms of the body; in particular, fever and swelling are involved in the reproduction of wound tissue by vasodilatation. Pain is also a part of the defense mechanism; however, the reaction to stressful psychological damage is more serious than the reaction that induces fever and swelling. Therefore, NSAIDs, such as loxoprofen (PGs’ inhibitor), are used to treat different inflammatory diseases. As the substance that specifically inhibits pain, IL-31 may be a more suitable analgesic drug for patients with various inflammatory diseases.

In the present study, the application of TNCB and capsaicin caused pain and decreased latency in the hot-plate test (45 °C) immediately or within 6 h after treatment. However, the onset of pain caused by the application of these materials was different. TNCB caused tissue injury and inflammation. Alternatively, the responses to noxious stimuli may be enhanced (hyperalgesia), or normally innocuous stimuli may produce pain (allodynia). PGs influence inflammation, and their administration induces the major signs of inflammation, including augmented pain [29,43]. However, the topical application of PGD_2_, PGE_2_, and PGI_2_ significantly suppressed LLS in skin-lesioned NC/Nga mice, and their inhibitory activities are in the order of PGD_2_ > PGI_2_ > PGE_2_ [28]. Moreover, PGs significantly enhanced nociceptive activity on hot-plate tests in mice [43]; PGD_2_, PGE_2_, and PGI_2_ significantly increased the nociceptive effect in the order of PGD_2_ > PGI_2_ > PGE_2_, which is the same as the order of their anti-pruritic activities. Although the scratching behavior strips off the epidermis and removes the alien substance invading the epidermis, it is a physiological reaction to directly control itching. In other words, scratching increases epidermal PGD_2_ biosynthesis, and PGD_2_-induced pain inhibits itching [27]. These previous and present findings suggest that cutaneous PGD_2_ could be mainly produced in response to pain and play a critical role in regulating the sensation of pain [44]. The significant individual differences in cutaneous IL-31 expression may be a reaction to TNCB-induced pain. Cutaneous IL-31 expression is unstable and intermittent; however, DRG neuronal IL-31RA expression always shows a stable increase through stimulation [5]. This phenomenon may be a physiological reaction to inhibit pain.

In the present study, capsaicin did not affect cutaneous PGD_2_ contents but significantly decreased DRG neuronal IL-31RA expression. These results indicate that the activation of TRPV1 and inhibition of IL-31RA mRNA expression in the DRG may mediate the antipruritic effects of capsaicin. The decrease in TRPV1 mRNA expression in the DRG triggered by capsaicin is hypothesized to respond to the onset of TRPV1-stimulated pain [30]. Therefore, we examined the effect of repeated administration of IL-31 on PGD_2_- or capsaicin-induced hyperalgesia on the modified hot-plate (45 °C) test. Repeated pretreatment with IL-31 inhibited PGD_2_-induced pain but not capsaicin-induced pain in the modified hot-plate test, suggesting that the site of action of IL-31 was after the PGs of the peripheral sensory nervous system. Furthermore, we assessed the effect of several analgesics, such as morphine, loxoprofen, and acetaminophen, on TNCB- or capsaicin-applied hot-plate (45 or 35 °C) tests to determine the effect of the IL-31-induced antinociceptive activity in wild-type and IL-31RAKI mice.

Morphine (3 mg/kg, subcutaneous) showed significant antinociception in the TNCB- or wild-type mice on the TNCB-applied hot-plate (45 °C) test but not in capsaicin- and IL-31RAKI mice. However, this was a partial inhibitory action as morphine-induced antinociceptive activity recovered in high doses (10 mg/kg, subcutaneously). Loxoprofen (15 mg/kg, oral) and acetaminophen (300 mg/kg, oral) showed significant antinociception on TNCB- or wild-type mice in the TNCB-applied hot-plate (35 °C) test; however, these antinociceptive activities were also completely inhibited by capsaicin- and IL-31RAKI mice. This result shows that the antinociceptive action of loxoprofen (NSAIDs) and acetaminophen does not develop without the activation of IL-31. Moreover, acetaminophen does not inhibit PG biosynthesis [45]. Although our present study suggested that the site of action of acetaminophen is the CNS, these collective results indicate that acetaminophen functions on the PNS, suggesting that IL-31 participates in the antinociceptive activity of these drugs. In addition, some antinociceptive activities remained in the morphine (opioid) group but not in the loxoprofen and acetaminophen groups. Chronic neuropathic pain shows strong resistance to treatment with opioids and NSAIDs [46]. This reaction is similar to drug action in chronic neuropathic pain. Therefore, the decrease in DRG neuronal IL-31RA may play a role in the onset of chronic neuropathic pain (Figure 10).

In the present study, we objectively evaluated the distinction between pain and itch using a wave pattern of scratching behavior by the movement of the hind leg of the mouse [21]. When the mouse skin is in a normal condition, e.g., histamine-injected intradermally, the mouse elicits 0.3–1.0 s of lasting scratching behavior (LLS), suggesting that the mouse felt something on the epidermis. When mice experience an itch, e.g., when IL-31 is injected intradermally, they elicit over 1.0 s lasting scratching behavior (LLS). This itching strips off the epidermis and removes an alien substance. For example, when a mite irritates the skin, IL-31 is highly expressed in that region, and an itch sensation is produced in response to various kinds of cutaneous stimulation [38]. If this stimulation continues for a long time, it may become atopic dermatitis due to the overexpression of DRG neuronal IL-31RA [39] (Figure 10). In contrast, when the mouse skin is experiencing a painful condition, e.g., TNCB (PGs inducer) application in the dorsal skin, the mouse elicits ultra-short-lasting scratching behavior (USLS) (less than 0.3 s) or no scratching behavior. Therefore, in the present study, we used a modified hot-plate test for a real pain evaluation. When the mouse was experiencing pain, it did not touch the skin, if possible. PGs is produced in response to inflammation, such as that in response to a burn, and may be involved in the onset of pain in response to various kinds of cutaneous stimulation. If this stimulation continues for a long time, it may induce chronic neuropathic pain due to the decreased expression of DRG neuronal IL-31RA (Figure 10).

In this study, we evaluated the effects of TNCB- or capsaicin-induced pain and itch after 24 h of this application on a modified hot-plate (45 °C) test. TNCB and capsaicin induced pain and inhibited itching after the application in a similar manner until 6 h, but they showed different reactions 24 h later. The TNCB application led to the development of pain torpor, which was not observed in IL-31RAKI mice. Therefore, this reaction was considered a result of endogenous IL-31. The itch (LLS counts) increase corresponded with the increase in the DRG neuronal IL-31RA expression. This reaction might be the cause of itching during the wound recovery period. The itch observed 24 h after TNCB application may be due to an increase in DRG neuronal IL-31RA expression after the restoration of the wound. In contrast, the application of capsaicin induced decreased latency 24 h after application and a significant decrease 6 days after. Strong pain is sometimes observed after the application of capsaicin cream when an individual takes a warm water shower. This pain can be explained by a decrease in DRG neuronal IL-31RA expression after capsaicin application. Although decreased latency caused by capsaicin was restored after 72 h of application, IL-31-induced (1 mg/kg, intravenously) LLS counts and DRG neuronal IL-31RA expression significantly decreased 144 h (6 days) after capsaicin application. IL-31RAKI mice lack a peripheral analgesic mechanism; therefore, the recovery from this latency involves the participation of the central analgesic mechanism.

These findings suggest that IL-31 may cause alloknesis; it alters the non-selective irritant stimulation into itch stimulation in mouse skin [47,48]. Therefore, it is possible that pain and itch are transmitted on the same nerve fibers, and a sensation is perceived as pain or itch depending on the operation of IL-31. However, PGD_2_ decreased LLS counts induced by the cutaneous injection of pruritogens or algogens in IL-31-induced itchy skin. These results indicate that PGD_2_ improves IL-31-induced alloknesis and it alters the non-selective irritant stimulation into pain stimulation in mouse skin [43]. Collectively, these data suggest that the sensation of pain and itch may be regulated by PGD_2_ (allodynia-inducer) [49] and/or IL-31 (alloknesis-inducer) [38], through their functional antagonism (Figure 10). 

The itch can be produced by a gentle touch, pressure, vibration, thermal, and electrical stimuli, such as transcutaneous or direct nerve stimulation. While itching may have different causes, it is conceived similarly by our senses. Our study indicates that the sensation of itch and pain depends on the type of stimulant of the peripheral sensory nerve ending. A strong pain will be felt, even if it is an itch if the burnt part is gently touched, suggesting that the sense of itch and pain are not stimulated from the outside, but from the inside. In this study, we were able to show that pain and itching sensations may be regulated by their functional antagonism as a phenomenon side, but the adjustment mechanism involved is still unknown and should be elucidated in future studies. And this study indicates that the analgesic action of IL-31 involves the PNS, which directly affects sensory nerves, providing a basis for developing novel analgesics using this mechanism. 

## 4. Materials and Methods

### 4.1. Animals

Male BALB/c and C57BL/6 mice aged 6–8 weeks were purchased from SLC Japan (Shizuoka, Japan). Mice lacking IL-31RA (IL-31RA^−/−^) were generated as described previously [50]. The IL-31RAKI (IL-31RA-deficient) mice used in this study were on a C57BL/6 genetic background and were obtained from hybrid mutant mice originally created based on a 129 SVJ-C57BL/6 background by backcrossing breeding over 15 generations. In this study, we used male homozygous (IL-31RAKI, IL-31RA^−/−^) and wild-type (IL-31RA^+/+^) mice matched for age. Therefore, we used C57BL/6 mice for the target group in the experiment using the IL-31RAKI mice. The animals were housed under a controlled temperature (23 ± 3 °C), humidity (50 ± 5%), and lighting (lights on from 7:00 am to 7:00 pm). All animals were provided free access to food and tap water. All procedures for animal experiments were approved by the Committee for Animal Experimentation at the International University of Health and Welfare following the Guidelines for Proper Conduct of Animal Experiments (Science Council of Japan, 2006).

### 4.2. Drugs

The mouse IL-31 cDNA-spanning amino acids 24–163 of IL-31 were cloned in a frame with pET30A (Novagen), and the construct was transformed into BL-21 cells (Novagen, Darmstadt, Germany). After induction with isopropyl-β-D-thiogalactopyranoside, IL-31 protein was purified under denaturation conditions using nickel-chelating sepharose (Qiagen, Benelux B.V., Hulsterweg, The Netherlands) and dialyzed in PBS [51]. IL-31, at a dose of 50 μg/kg, was used for subcutaneous or intravenous injection, as previously described [39]. Morphine hydrochloride (Takeda Pharmaceutical Co., Ltd., Osaka, Japan) was dissolved in saline and administered at 0.5 mL/kg, subcutaneously. Loxoprofen (Loxonin, Daiichi-Sankyou, Tokyo, Japan) and acetaminophen (Tylenol, Towa Pharmaceutical Co., Ltd., Toyama, Japan) were purchased as commercially available drugs. They were suspended in 0.3% carboxymethyl cellulose sodium saline (CMC), 0.5 mL/kg, for oral administration. TNCB (2,4,6-trinitrochlorobenzene, Tokyo Kasei, Tokyo, Japan) or capsaicin (Wako Jyunyaku, Tokyo, Japan) were dissolved in an acetone ethanol mixture (acetone: ethanol = 1:4, AEM) or ethanol and applied to the limbs of mice (0.04 mL/site) using a pipette. The application of 0.16 mL TNCB and capsaicin on the hot-plate test was sufficient for this method. Aliquots of 0.04 mL/site were applied to the site of application. Scratching counts can be measured by evaluating the expression of several mRNAs, the measurement of cutaneous PGD_2_ contents tests, and the application of 0.2 mL TNCB or capsaicin/site using a pipette.

### 4.3. Hot-Plate Test

The hot plate test [12] was used to measure withdrawal latency as described previously [19]. The hot-plate test was improved to evaluate even weak analgesic drugs such as NSAIDs: loxoprofen. Mice were placed on a hot-plate maintained at 50 ± 0.5 °C, and the latency to either paw-lick or an attempt to escape by jumping was recorded. To prevent tissue damage, mice that showed no response within 60 s were removed from the hot-plate and assigned a score of 60 s. The percentage of nociception (nociceptive index) was calculated according to the formula: [(*T*_1_ − *T*_0_)/(*T*_2_ − *T*_0_)] × 100, where *T_0_* and *T_1_* were the latencies observed before and after the drug administration, respectively, and *T_2_* was the cut-off time (60 s). The modified hot-plate test applied 2,4,6-trinitrochlorobenzene (TNCB, 0.3, 1, and 3 *w*/*v* %) to the limbs of mice at a lower temperature (30 or 45 °C) than that used in the hot-plate test. Animals were tested before and 30, 60, 90, 120, 150, and 180 min after drug administration. As the number of experiments in which pretreatment of TNCB, capsaicin, or PGD_2_ application was over the cut-off level of 60 s was high, it was impossible to calculate many nociceptive indices at 35 or 45 °C. Therefore, the data were transcribed in latency.

### 4.4. Measurement of Scratching Counts

Scratching counts were measured as previously described [43]. A small magnet (1.0 mm diameter, 3.0 mm length) was implanted subcutaneously into both hind paws of isoflurane-anesthetized mice 24 h before the measurement. Each mouse was placed in an observation chamber (11 cm diameter, 18 cm height) surrounded by a circular coil, through which electric current, induced by movement of the magnets attached to the hind paws, was amplified and recorded. The number of spontaneous scratches was automatically detected and objectively evaluated via a computer using MicroAct (Neuroscience, Tokyo, Japan) [52]. The analysis parameters for detecting waves were as follows: threshold, 0.1 V; event gap, 0.2 s; minimum duration, 0.3 or 1.0 s; maximum frequency, 20 Hz; and minimum frequency, 2 Hz.

### 4.5. Measurement of Dorsal Cutaneous Temperature and Skin Weight

Time-course changes of dorsal cutaneous surface temperature were measured after 0 (pre), 1-, 3-, 6-, 24-, and 72 h after 3 *w*/*v*% TNCB application in the dorsal skin of mice. The dorsal hair of the mouse was shaved off with an electric razor, and 0.2 mL of 3% TNCB solution was applied using a pipette. The cutaneous surface temperature was measured using a non-contact type thermometer (HPC-01, Harasawa Co. Ltd., Tokyo, Japan). Mice were killed by dislocation of the cervical vertebrae after 6, 24, and 36 h of TNCB application. The dorsal skin of each mouse was removed with a fixed circular area (Φ 1.6 mm) at the site of the TNCB application. The skin was weighed, and the oncotic index of skin swelling was measured. 

### 4.6. Measurement of Cutaneous Prostaglandin D_2_ Content

The mice were injected intravenously with indomethacin (10 mg/kg) 1 h after TNCB application, as well as 1, 3, 6, and 12 days after TNCB application to prevent further production of prostaglandins D_2_ (PGD_2_). Five minutes after the indomethacin injection, the mice were killed by dislocation of the cervical vertebrae, and approximately 100 mg of the back skin of each mouse was removed. The skin was minced and homogenized in ice-cold phosphate-buffered saline containing 0.1 mM indomethacin with a Polytron tissue homogenizer for 30 s on ice. Four milliliters of acetone were added to the sample and mixed. The precipitate was then removed by centrifugation at 2000× *g* for 10 min at 4 °C. The supernatant was carefully poured into a test tube, evaporated under a stream of nitrogen, and re-suspended in an enzyme immunoassay (EIA) buffer. The amount of PGD_2_ in the suspension was measured using specific EIA kits (Cayman Chemical, Ann Arbor, MI, USA) following the manufacturer’s instructions.

### 4.7. Real-Time Quantitative PCR

The gene expression levels of IL-31, IL-31RA, TRPV1, and β-actin were measured using real-time polymerase chain reaction (RT-PCR) in the DRG (C_4–7_, T_1–4_) neuron cell body from the shoulder and back of the BALB/c and C57BL/6 mice at each point. Total RNA was extracted from the dorsal skin of each mouse by Trizol (Invitrogen, Carlsbad, CA, USA) and digested using amplification-grade DNase I (Invitrogen), according to the manufacturer’s instruction. cDNA was synthesized by the SuperScript III First-Strand Synthesis System (Invitrogen). Quantitative RT-PCR was performed with SYBR Green Master Mix, using an Applied Biosystems 7700 Sequence Detection System (Applied Biosystems, Foster City, CA, USA). The PCR primers for IL-31 were designed using PRIMER 3 software (v. 0.4.0), and primers for TRPV1 and β-actin were purchased from TAKARA BIO (Otsu, Shiga, Japan). Primer sequences were as follows: IL-31 (5′-ATA CAG CTG CCG TGT TTC AG-3′ and 5′ -AGC CAT CTT ATC ACC CAA GAA-3′), IL-31RA (5′-CCA GAA GCT GCC ATG TCG AA-3′ and 5′-TCT CCA ACT CGG TGT CCC AAC-3′), TRPV1 (5′-CAA CAA GAA GGG GCT TAC ACC-3′ and 5′ -TCT GGA GAA TGT AGG CCA AGA C-3′), and β—actin (5′ -TGA CAG GAT GCA GAA GGA GA-3′ and 5′-GCT GGA AGG TGG ACA GTG AG-3′). Relative expression levels were calculated using the relative standard curve method as outlined in the manufacturer’s technical bulletin. A standard curve was generated using the fluorescence data obtained from four-fold serial dilutions of the total RNA of the sample with the highest expression. The curve was then used to calculate the relative amounts of target mRNA in test samples. Quantities of all targets in the test samples were normalized to the corresponding β-actin RNA transcript in skin samples.

### 4.8. Statistical Analysis

All data were statistically analyzed using GraphPad InStat and GraphPad Prism (GraphPad Software, Version 7, Inc., La Jolla, CA, USA). Experimental values are expressed as means and standard errors (S.E.). Data on time-course changes in scratching counts or the percentage of mRNA expression were analyzed using one-way ANOVA followed by the Student–Newman–Keuls test. Then, other data were analyzed using one-way, two-way, or three-way ANOVA followed by Bonferroni or Tukey tests. In some cases, when a main effect was significant without interaction effect, we did an exploratory and limited pairwise post hoc comparison consistent with our a priori hypothesis. *p*-values less than 0.05 were considered statistically significant.

## 5. Conclusions

Itch and pain are common senses caused by several drugs and physical stimulations. However, the onset mechanisms of itch and pain remain unclear. We objectively evaluated itch and pain via a wave pattern of scratching behavior in mice. Scratching behavior in mice is divided into two types, LLS (itch-associated scratching) and SLS (hygiene behavior). Moreover, morphine is one of the few drugs that induce LLS. In IL-31RAKI mice, LLS disappeared upon the administration of morphine, accompanied by the partial disappearance of antinociceptive action. Moreover, we found that IL-31 was partially involved in the peripheral analgesic mechanism and that IL-31-induced alloknesis inhibited pain. The antinociceptive activity of IL-31 was approximately as effective as that of loxoprofen (NSAIDs). However, the site of action of IL-31 was in the peripheral sensory nerve after the PGs biosynthesis inhibition, unlike NSAIDs. In addition, cutaneous IL-31 expression increases at the onset of pain and DRG neuronal IL-31RA expression increased during recovery. Our results suggest that IL-31 functions via a physiological pain-inhibitory mechanism. These results indicate that the analgesic action of IL-31 involves the peripheral nervous system, which affects sensory nerves and provides a basis for the development of novel analgesics that utilize this mechanism.

## Figures and Tables

**Figure 1 ijms-24-11563-f001:**
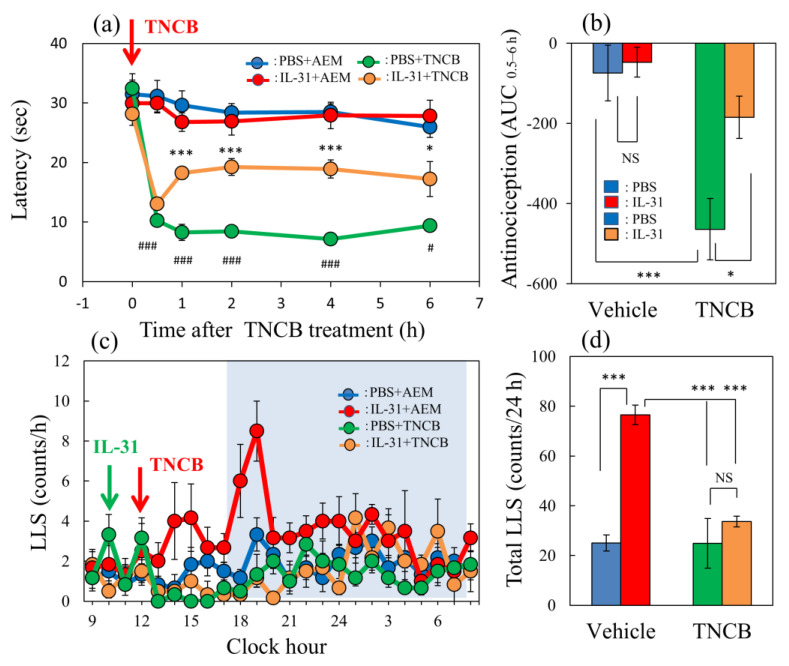
Effect of IL-31 on TNCB-applied hot-plate test in BALB/c mice. (**a**) Effect of repeated pretreatment of IL-31 (50 μg/kg, intraperitoneally, every 12 h for three days) on TNCB (3%, 0.04 mL/each limb, total 0.16 mL/site)-applied hot-plate (45 °C) test. ^#^ *p* < 0.05, ^###^ *p* < 0.001 compared with the corresponding values in the saline + vehicle-treated group. (**b**) Total antinociceptive index of IL-31 after 0.5 to 6 h (AUC_0.5–6 h_). (**c**) Time-course change of itch-associated scratching behavior (LLS, counts/h) after the application of TNCB (3%, 0.2 mL/dorsal site). The blue line indicates the PBS + AEM-treated group; the red line indicates IL-31 + AEM-treated group; the green line indicates PBS + TNCB-treated group; the yellow line indicates IL-31 + TNCB-treated group. The blue column indicates the PBS + AEM-treated group; the red column indicates the IL-31 + AEM-treated group; the green column indicates the PBS + TNCB-treated group; the yellow column indicates the IL-31 + TNCB-treated group. (**a**,**b**) * *p* < 0.05, *** *p* < 0.001 compared with the corresponding values in the PBS + TNCB-treated group. The lateral axis indicates the clock hour, and the shaded area represents nighttime (dark phase, 7:00 pm to 7:00 am). (**d**) Total LLS counts for 24 h. The blue column indicates the IL-31 + AEM-treated group; the red column indicates the IL-31 + TNCB-treated group; the green column indicates the PBS + TNCB-treated group; the yellow column indicates the IL-31 + TNCB-treated group. The red arrow indicates the AEM or TNCB application point. The blue arrow indicates the IL-31 (1 mg/kg, intraperitoneally) administration point. Each value represents the mean ± standard error (S.E.) from 6 mice (total 36 mice). *** *p* < 0.001 compared with the vehicle (PBS) + TNCB-treated group. Three-way ANOVA: group × treatment × time: F (5, 120) = 2.760, *p* = 0.0214, (**a**); group × treatment × time: F (23, 480) = 1.230, *p* = 0.2124, (**c**). Two-way ANOVA: group × treatment: F (1, 20) = 4.303, *p* = 0.0512, group: F (1, 20) = 18.68, *p* = 0.0003, treatment: F (1, 20) = 6.296, *p* = 0.0208, (**b**); group × treatment: F (1, 20) = 13.95, *p* = 0.0013, group: F (1, 20) = 14.16, *p* = 0.0012, treatment: F (1, 20) = 27.89, *p* < 0.0001, (**d**).

**Figure 2 ijms-24-11563-f002:**
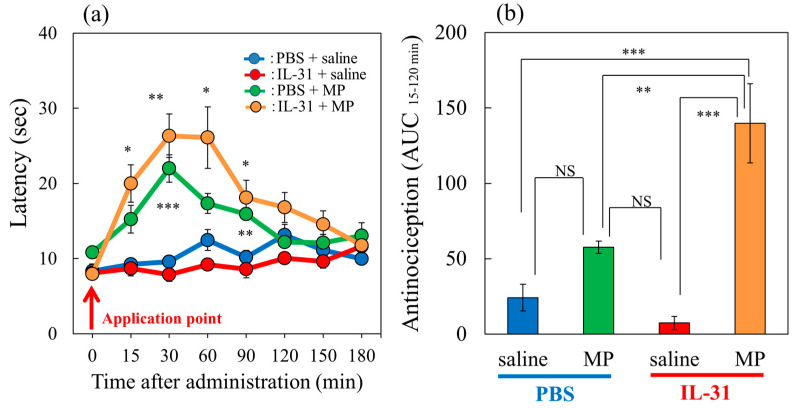
Effects of IL-31 and morphine on conventional hot-plate (51 °C) test in BALB/c mice. (**a**) Effect of repeated pretreatment of IL-31 (50 μg/kg, intraperitoneally, every 12 h for three days) and a single dose of morphine (MP, 3 mg/kg, subcutaneously) on conventional hot-plate (51 °C) test. The blue line indicates the phosphate-buffered saline (PBS) + saline-treated group; the red line indicates the IL-31 + saline-treated group; the green line indicates PBS + morphine-treated group; the yellow line indicates IL-31 + morphine-treated group. Each value represents the mean ± standard error (S.E.) from 6 mice (24 mice). * *p* < 0.05, ** *p* < 0.01, *** *p* < 0.001 compared with the corresponding values in the PBS + saline-treated group. (**b**) Total antinociceptive index of morphine after 15 to 120 min (AUC_15–120 min_). The blue column indicates PBS + saline-treated group; the green column indicates PBS + morphine-treated group. The red column indicates the IL-31 + saline-treated group; the yellow column indicates IL-31 + morphine-treated group. Each value represents the mean ± standard error (S.E.) from 6 mice (total 24 mice). * *p* < 0.05, ** *p* < 0.01, *** *p* < 0.001 compared with each group. Three-way ANOVA: group × treatment × time: F (7, 168) = 2.360, *p* = 0.0252, (**a**). Two-way ANOVA: group × treatment: F (1, 20) = 12.11, *p* = 0.0024, group: F (1, 20) = 34.12, *p* < 0.0001, treatment: F (1, 20) = 5.28, *p* = 0.0325, (**b**).

**Figure 3 ijms-24-11563-f003:**
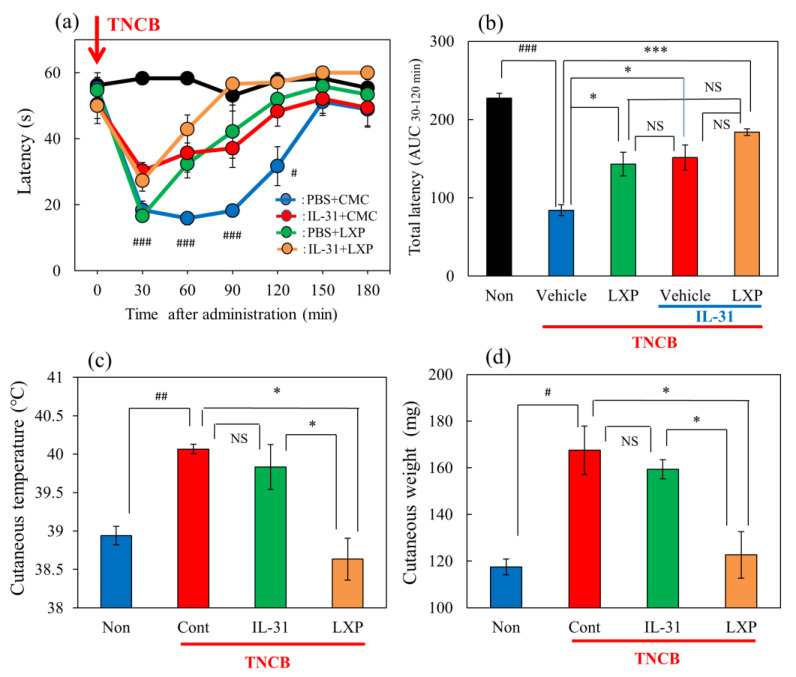
Effects of IL-31 and loxoprofen (LXP) on TNCB–applied the hot-plate test in BALB/c mice. (**a**) Effects of IL-31 and loxoprofen (LXP) on TNCB-applied hot-plate (45 °C) test in mice. The black line and column indicate the non-TNCB-treated group; the blue line indicates phosphate-buffered saline (PBS) + vehicle (carboxymethyl cellulose sodium, CMC, 10 mL/kg, oral)-treated group; the red line indicates IL-31 + CMC-treated group; the green line indicates PBS + LXP-treated group; the yellow line indicate IL-31 + LXP treated group. The red arrow indicates the AEM or TNCB application point. ^#^
*p* < 0.05, ^###^ *p* < 0.001 compared with the corresponding values in the saline + vehicle-treated group. (**b**) Total antinociceptive index of LXP after 30–120 min (AUC_30–120 min_). The black line and column indicate the non-treated group; the blue column indicates the PBS + CMC-treated group; the green column indicates the PBS + LXP-treated group; the red column indicates the IL-31 + CMC treated-group; the yellow column indicates the IL-31 + LXP-treated group. Each value represents the mean ± standard error (S.E.) from 6 mice (total 54 mice). NS, not significant, ^###^ *p* < 0.001 compared with the corresponding values in the saline + vehicle-treated group. * *p* < 0.05, *** *p* < 0.001 compared with the corresponding values in the vehicle + TNCB treated group; (**c**) Effects of IL-31 and LXP on TNCB-induced increasing cutaneous temperature (fever) after 24 h application of TNCB. (**d**) Effects of IL-31 and LXP on TNCB-induced increasing cutaneous weight (swelling) after 24 h application of TNCB. The blue column indicates saline + vehicle-treated group; the red column indicates vehicle + TNCB-treated group; the green column indicates IL-31 + TNCB-treated group; the yellow column indicates LXP + TNCB-treated group. (**c**,**d**) Each value represents the mean ± S.E. from 6 mice (total 24 mice), NS, not significant, ^#^ *p* < 0.05, ^##^
*p* < 0.01 compared with the non-TNCB-treated group. * *p < 0.05* when compared with each group. Two-way ANOVA: time × group: F (24, 175) = 5.382, *p* < 0.0001, time: F (6, 175) = 34.22, *p* < 0.0001, group: F (4, 175) = 37.29, *p* < 0.0001, (**a**). One-way ANOVA: F (4, 25) = 111, *p* < 0.0001, (**b**); F (3, 20) = 10.65, *p* = 0.0002, (**c**); F (3, 20) = 10.87, *p* = 0.0002, (**d**).

**Figure 4 ijms-24-11563-f004:**
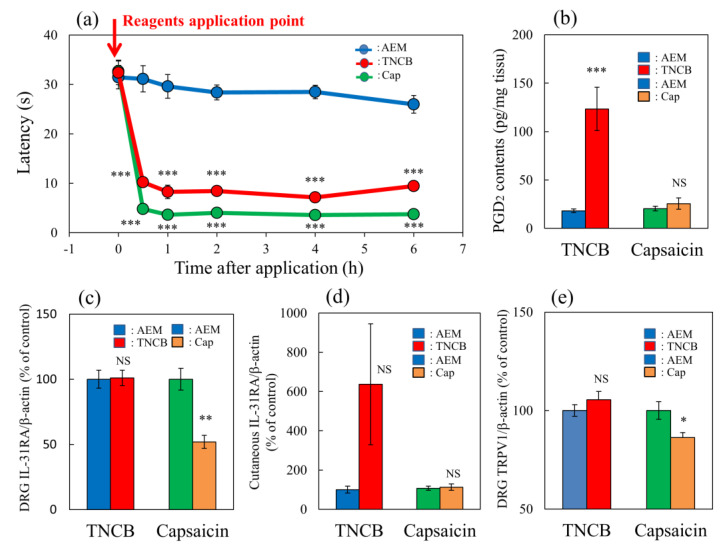
Comparative effects of TNCB or capsaicin on latency in modified hot-plate test and several pain-related parameters in BALB/c mice. (**a**) Time-course changes of latency caused by topical application of TNCB (3%) or capsaicin (Cap, 1%) in modified hot-plate (45 °C) test. The red arrow indicates the vehicle (acetone-ethanol mixed liquor, AEM, 0.2 mL/site) or TNCB or capsaicin application point. The values represent the means ± standard error (S.E.) from 6 mice. *** *p* < 0.001 compared with the respective value of the AEM-treated group. (**b**) Effects of TNCB and capsaicin on cutaneous PGD_2_ contents 1 h after application. (**c**) Effects of TNCB or capsaicin on DRG neuronal IL-31RA mRNA expression 6 h after application. (**d**) Effects of TNCB or capsaicin on cutaneous IL-31 mRNA expression 1 h after application. € Effects of TNCB or capsaicin on DRG neuronal TRPV1 mRNA expression 6 h after application. The values represent the means ± standard error (S.E.) from 6 mice (total 42 mice). (**a**–**d**) NS, not significant, * *p* < 0.05, ** *p* < 0.01, *** *p* < 0.001 compared with the respective value of the AEM-treated group. Two-way ANOVA: time × group: F (10, 90) = 14.22, *p* < 0.0001, time: F (5, 90) = 74.75, *p* < 0.0001, group: F (2, 90) = 307.4, *p* < 0.0001, (**a**); group × treatment: F (1, 20) = 18.51, *p* = 0.0003, group: F (1, 20) = 22.51, *p* = 0.0001, treatment: F (1, 20) = 16.9, *p* = 0.0005, (**b**); group × treatment: F (1, 20) = 10.65, *p* = 0.0039, group: F (1, 20) = 9.817, *p* = 0.0052, treatment: F (1, 20) = 10.65, *p* = 0.0039, (**c**); group × treatment: F (1, 20) = 3.016, *p* = 0.0978, group: F (1, 20) = 3.014, *p* = 0.0979, treatment: F (1, 20) = 3.016, *p* = 0.0978, (**d**); group × treatment: F (1, 20) = 6.93, *p* = 0.0160, group: F (1, 20) = 1.273, *p* = 0.2725, treatment: F (1, 20) = 6.932, *p* = 0.0159, (**e**).

**Figure 5 ijms-24-11563-f005:**
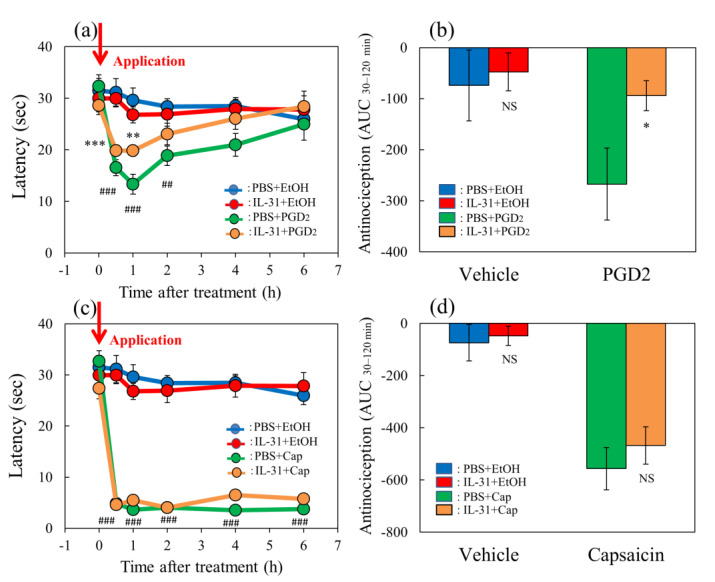
Effects of IL-31 on PGD_2_- or capsaicin-applied hot-plate test in BALB/c mice. (**a**) Effect of IL-31 on PGD_2_-induced decreasing latency on hot–plate (45 °C) test. ** *p* < 0.01, *** *p* < 0.001 compared with the corresponding values in the vehicle (ethanol, EtOH, 0.2 mL/site)-treated group; ## *p* < 0.01, ### *p* < 0.001 compared with the corresponding values in the vehicle-treated group. (**b**) Total antinociceptive index from 30 to 120 min (AUC_30–120 min_) after PGD_2_ application. NS, not significant, * *p* < 0.05 compared with the corresponding values in the vehicle-treated group. (**c**) Effect of IL-31 on capsaicin (Cap)-induced decreasing latency on modified hot-plate (45 °C) test. ### *p* < 0.001 compared with the corresponding values in the vehicle-treated group. (**d**) After capsaicin application, the total antinociceptive index changes from 30 to 120 min (AUC_30–120 min_). The blue line indicates the vehicle (phosphate-buffered saline, PBS, intraperitoneal, every 12 h for 3 days) + EtOH-group; the red line indicates the IL-31-pretreated (50 μg/kg, intraperitoneal, every 12 h for 3 days) + EtOH-treated group; the green line indicates PBS + PGD_2_ (0.1%, 0.04 mL/each limb)- or capsaicin (3%, 0.04 mL/each limb)-treated group; the yellow line indicates IL-31 + PGD_2_- or capsaicin-treated group. The red arrow indicates the vehicle (ethanol, EtOH)- or PGD_2_ or capsaicin application point. Values represent the mean ± S.E. from 6 mice (total of 48 mice). NS, not significant, compared with the respective values of the vehicle-treated group. Three-way ANOVA: group × treatment × time: F (5, 120) = 0.885, *p* = 0.4934, group × treatment: F (1, 120) = 5.631, *p* = 0.0192, group: F (1, 120) = 47.782, *p* < 0.0001, treatment: F (1, 120) = 1.591, *p* = 0.2096, time: F (5, 120) = 7.123, *p* < 0.0001, (**a**); group × treatment × time: F (5, 126) = 0.824, *p* = 0.5347, group: F (1, 126) = 903.432, *p* < 0.0001, treatment: F (1, 126) = 0.315, *p* = 0.5759, time: F (5, 126) = 50.026, *p* < 0.0001, (**c**). Two-way ANOVA: group × treatment: F (1, 20) = 1.79, *p* = 0.1959, group: F (1, 20) = 3.317, *p* = 0.0835, treatment: F (1, 20) = 4.791, *p* = 0.0406, (**b**); group × treatment: F (1, 21) = 0.2122, *p* = 0.6498, group: F (1, 21) = 44.57, *p* < 0.0001, treatment: F (1, 21) = 0.727, *p* = 0.4035, (**d**).

**Figure 6 ijms-24-11563-f006:**
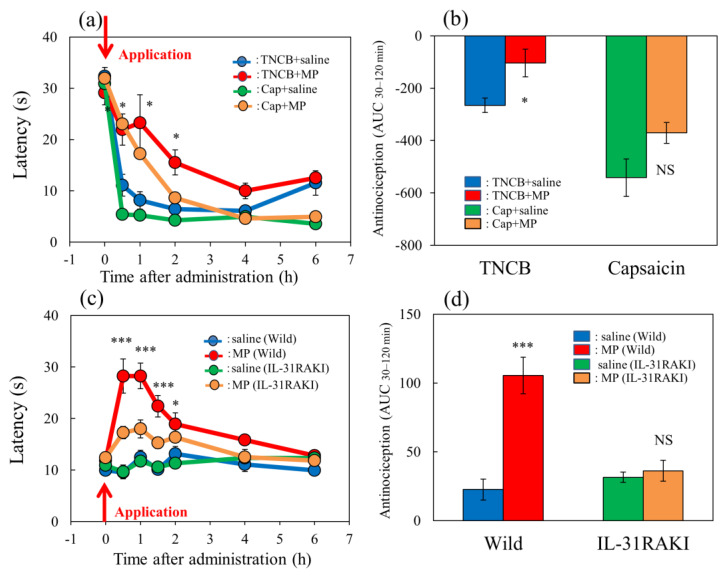
Effect of morphine on TNCB- or capsaicin-applied hot-plate test in wild-type or IL-31RAKI mice. (**a**) Effect of morphine (3 mg/kg, subcutaneous) on TNCB- or capsaicin-applied hot-plate (45 °C) test in BALB/c mice. Each value represents the mean ± standard error (S.E.) from 6 mice * *p* < 0.05 compared with the respective values in vehicle (saline)-treated mice. (**b**) Total antinociceptive index of morphine after 30 to 120 min (AUC_30–120 min_) as per the TNCB or capsaicin-treated hot-plate test. NS, not significant, * *p* < 0.05 compared with the respective values in vehicle-treated mice. The blue line and column indicate the TNCB + vehicle (saline)-treated group; the red line and column indicates the TNCB + morphine-treated group; the green line and column indicates the capsaicin + saline-treated group; the yellow line and column indicates the capsaicin + morphine-treated group. The red arrow indicates the TNCB or capsaicin application point. (**c**) Effect of morphine on conventional hot-plate (51 °C) test in wild-type (C57BL/6) and IL-31RAKI (C57BL/6 genetic background) mice. * *p* < 0.05, and *** *p* < 0.001 compared with the respective values in vehicle-treated mice. (**d**) Total antinociceptive index of morphine after 30 to 120 min (AUC_30–120 min_) of wild-type and IL-31RAKI mice. The blue line and column indicate the vehicle (saline)-treated group in wild-type mice; the red line and column indicates the morphine-treated group in wild-type mice; the green line and column indicates the saline-treated group in IL-31RAKI mice; the yellow line and column indicates the morphine-treated group in IL-31RAKI mice. Wild, C57BL/6 mice; IL-31RAKI, IL-31 receptor A-deficient mice (C57BL/6 genetic background). Each value represents the mean ± standard error (S.E.) from 6 mice (total 48 mice). *** *p* < 0.001 compared with the respective values in vehicle-treated mice. Three-way ANOVA: group × treatment × time: F (5, 120) = 1.275, *p* = 0.2792, group: F (1, 120) = 17.296, *p* < 0.0001, treatment: F (1, 120) = 49.027, *p* < 0.0001, time: F (5, 120) = 73.243, *p* < 0.0001, (**a**); group × treatment × time: F (7, 160) = 2.171, *p* = 0.0395, (**c**). Two-way ANOVA: group × treatment: F (1, 20) = 0.007463, *p* = 0.9320, group: F (1, 20) = 28.95, *p* < 0.0001, treatment: F (1, 20) = 10.85, *p* = 0.0036, (**b**); group × treatment: F (1, 20) = 19.79, *p* = 0.0002, group: F (1, 20) = 11.82, *p* = 0.0026, treatment: F (1, 20) = 24.87, *p* < 0.0001, (**d**).

**Figure 7 ijms-24-11563-f007:**
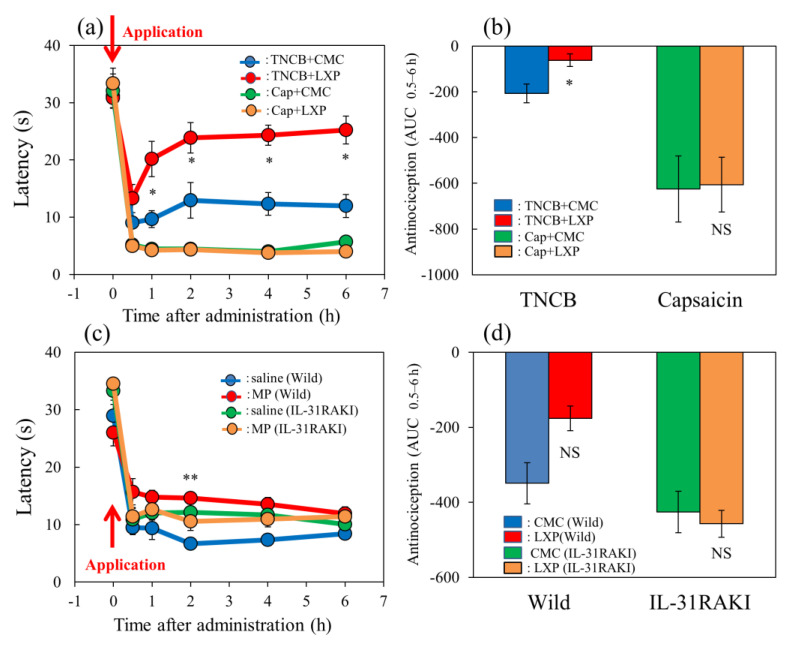
Effect of loxoprofen on TNCB- or capsaicin-applied hot-plate test in wild-type and IL-31RAKI mice. (**a**) Effect of loxoprofen on TNCB- or capsaicin-applied hot-plate (45 °C) test in BALB/c mice. Each value represents the mean ± standard error (S.E.). * *p* < 0.05 compared with the respective values in the vehicle (carboxymethyl cellulose sodium saline, CMC, 10 mL/kg, oral)-treated mice. (**b**) Total antinociceptive index of loxoprofen after 0.5 to 6 h (AUC_0.5–6 h_) as per the TNCB- or capsaicin-applied hot-plate test in BALB/c mice. Each value represents the mean ± standard error (S.E.) from 6 mice. NS, not significant, * *p* < 0.05 compared with the respective values in the CMC-treated mice. The blue line and column indicate the TNCB + vehicle (saline)-treated group; the red line and column indicates the TNCB + loxoprofen-treated group; the green line and column indicates the capsaicin + saline-treated group; the yellow line and column indicates the capsaicin + loxoprofen-treated group. (**c**) Effect of loxoprofen on TNCB-applied hot-plate (45 °C) test in wild-type (C57BL/6) and IL-31RAKI (C57BL/6 genetic background) mice. Each value represents the mean ± standard error (S.E.). ** *p* < 0.01 compared with the respective values in the CMC-treated mice. (**d**) Total antinociceptive index of loxoprofen (AUC_0.5–6 h_) of wild-type and IL-31RAKI mice. NS, not significant, compared with the respective values in vehicle-treated mice. The blue line indicates TNCB + CMC-and column indicate the vehicle (saline)-treated group in wild-type mice; the red line and column indicates TNCB +the loxoprofen-treated (15 mg/kg, oral) group in wild-type mice; the green line and column indicates capsaicin + CMCthe saline-treated group in IL-31RAKI mice; the yellow line and column indicates capsaicin +the loxoprofen -treated group in IL-31RAKI mice. The red arrow indicates the TNCB or capsaicin application point. Wild, C57BL/6 mice; IL-31RAKI, IL-31 receptor A-deficient mice (C57BL/6 genetic background). Each value represents the mean ± standard error (S.E.) from 6 mice (total 48 mice). Three-way ANOVA: group × treatment × time: F (5, 120) = 2.430, *p* = 0.0389, (**a**); group × treatment × time: F (5, 96) = 1.095, *p* = 0.3685, group: F (1, 96) = 6.646, *p* = 0.0115, treatment: F (1, 96) = 14.570, *p* = 0.0002, time: F (5, 96) = 102.550, *p* < 0.0001, (**c**). Two-way ANOVA: group × treatment: F (1, 20) = 0.4223, *p* = 0.5232, group: F (1, 20) = 24.67, *p* < 0.0001, treatment: F (1, 20) = 0.7201, *p* = 0.4062, (**b**); group × treatment: F (1, 16) = 4.854, *p* = 0.0426, group: F (1, 16) = 10.21, *p* = 0.0056, treatment: F (1, 16) = 3.69, *p* = 0.0727, (**d**).

**Figure 8 ijms-24-11563-f008:**
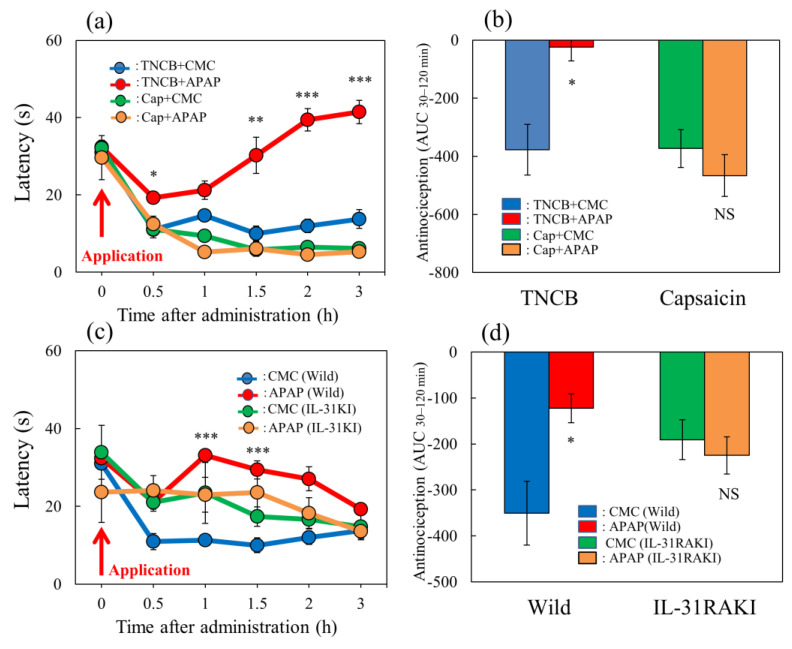
Effect of acetaminophen on TNCB– or capsaicin–applied hot-plate (35 °C) test in wild-type and IL-31RAKI mice. (**a**) Effect of acetaminophen (APAP, 300 mg/kg, oral) on TNCB- or capsaicin-applied hot-plate (35 °C) test in BALB/c mice. Each value represents the mean ± standard error (S.E.).* *p* < 0.05, ** *p* < 0.01, and *** *p* < 0.001 compared with the respective values in vehicle (carboxymethyl cellulose sodium, CMC, 10 mL/kg, oral)-treated mice. (**b**) Total antinociceptive index of acetaminophen after 30–120 min (AUC_30–120 min_) as per the TNCB- or capsaicin-applied hot-plate test in BALB/c mice. Each value represents the mean ± standard error (S.E.). NS, not significant, * *p* < 0.05 compared with the respective values in vehicle-treated mice. The blue line and column indicate the TNCB + vehicle (saline)-treated group; the red line and column indicates the TNCB + acetaminophen-treated group; the green line and column indicates the capsaicin + saline-treated group; the yellow line and column indicates the capsaicin + acetaminophen-treated group. (**c**) Effect of acetaminophen on TNCB-applied hot-plate (35 °C) test in wild-type (C57BL/6) mice and IL-31RAKI (C57BL/6 genetic background) mice. Each value represents the mean ± standard error (S.E.). *** *p* < 0.001 compared with the respective values in the CMC-treated mice. (**d**) Total antinociceptive index of acetaminophen after 30–120 min (AUC_30–120 min_) of wild-type and IL-31RAKI mice. The blue line indicatesand column indicate the TNCB + CMC-vehicle (saline)-treated group in wild-type mice; the red line and column indicates the TNCB + acetaminophenAPAP-treated group in wild-type mice; the green line and column indicates the capsaicin + CMCsaline-treated group in IL-31RAKI mice; the yellow line and column indicates the capsaicin + acetaminophenAPAP-treated group in IL-31RAKI mice. The red arrow indicates the TNCB or capsaicin application point. Wild, C57BL/6 mice; IL-31RAKI, IL-31 receptor A-deficient mice (C57BL/6 genetic background). Each value represents the mean ± standard error (S.E.) from 6 mice (total 48 mice). NS, not significant, * *p* < 0.05 compared with the respective values in the CMC-treated mice. Three-way ANOVA: group × treatment × time: F (5, 126) = 1.200, *p* = 0.3128, group: F (1, 126) = 381.730, *p* < 0.0001, treatment: F (1, 126) = 6.316, *p* = 0.0132, time: F (5, 126) = 44.926, *p* < 0.0001, (**a**); group × treatment × time: F (5, 120) = 0.248, *p* = 0.9400, group: F (1, 120) = 2.609, *p* = 0.1089, treatment: F (1, 120) = 21.475, *p* < 0.0001, time: F (5, 120) = 4.594, *p* = 0.0007, (**c**). Two-way ANOVA: group × treatment: F (1, 20) = 10.33, *p* = 0.0043, group: F (1, 20) = 9.935, *p* = 0.0050, treatment: F (1, 20) = 3.516, *p* = 0.0754, (**b**); group × treatment: F (1, 20) = 10.33, *p* = 0.0043, group: F (1, 20) = 9.935, *p* = 0.0050, treatment: F (1, 20) = 3.516, *p* = 0.0754, (**d**).

**Figure 9 ijms-24-11563-f009:**
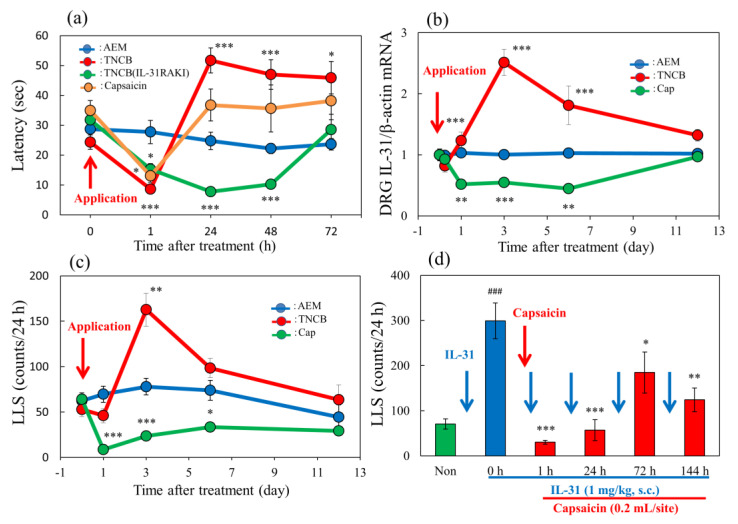
Long–term changes after TNCB or capsaicin application on the sense of itch and pain. (**a**) Effects of topical application of TNCB or capsaicin on modified hot-plate (45 °C) test in BALB/c mice and IL-31RAKI mice. The blue line indicates the vehicle (acetone-ethanol mixed liquor, AEM)-treated mice; the red line indicates the TNCB-treated mice; the green line indicates the capsaicin-treated mice; the yellow line indicates the TNCB-treated IL-31RAKI mice. (**b**) Effects of topical application of TNCB or capsaicin on IL-31RA expression in the DRG. The blue line indicates AEM-treated mice; the red line indicates TNCB-treated mice; the green line indicates capsaicin-treated mice. (**c**) Effects of topical application of TNCB or capsaicin on itch-associated behavior counts (LLS counts/24 h). The blue line indicates AEM-treated mice; the red line indicates TNCB-treated mice; the green line indicates capsaicin-treated mice. The red arrow indicates the AEM or TNCB or capsaicin application point. Each value represents the mean ± standard error (S.E.) from 6 mice (total 54 mice). * *p* < 0.05, ** *p* < 0.01, and *** *p* < 0.001 compared with the respective values of vehicle (AEM)-treated group. (**d**) Time-course change of topical application of capsaicin on IL-31-induced LLS counts. The blue arrows indicate the IL-31 (1 mg/kg, intravenous) administration point. The red arrow indicates the capsaicin (1%, 0.2 mL/site) application point. Each value represents the mean ± standard error (S.E.) from 6 mice. * *p* < 0.05, ** *p* < 0.01, and *** *p* < 0.001 compared with IL-31-induced LLS counts before capsaicin application. Two-way ANOVA: group × time: F (12, 100) = 7.899, *p* < 0.0001, group: F (3, 100) = 18.81, *p* < 0.0001, time: F (4, 100) = 12.45, *p* < 0.0001, (**a**); group × time: F (10, 90) = 15.2, *p* < 0.0001, group: F (2, 90) = 63.57, *p* < 0.0001, time: F (5, 90) = 6.516, *p* < 0.0001, (**b**); group × time: F (8, 75) = 8.204, *p* < 0.0001, group: F (2, 75) = 31.92, *p* < 0.0001, time: F (4, 75) = 9.405, *p* < 0.0001, (**c**). One-way ANOVA: F (5, 30) = 12.11, *p* < 0.0001, (**d**).

**Figure 10 ijms-24-11563-f010:**
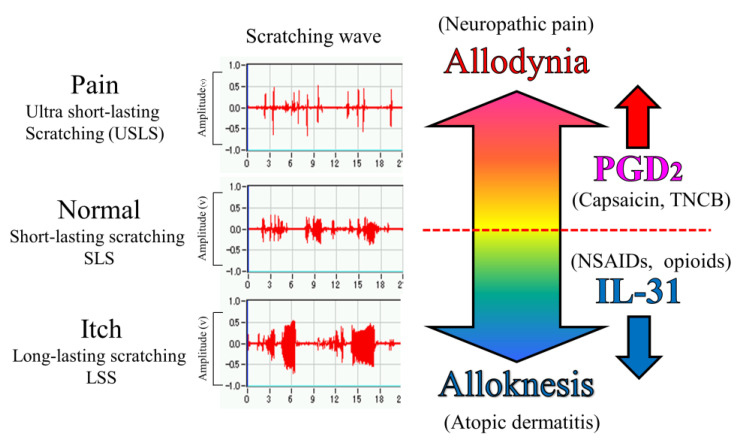
Schematic diagram of the putative roles of IL-31 and PGD_2_ in the regulation of the sensation of cutaneous itch and pain in mice. We evaluated a wave pattern of scratching behavior from the movement of the hind leg of the mouse, objectively. In normal conditions, the mouse elicits 0.3–1.0 s lasting scratching behavior (SLS) caused by several stimulants. Under itchy conditions, the mouse elicits over 1.0 s lasting scratching behavior (LLS) caused by IL-31. If this stimulation continues for a long time, it may become atopic dermatitis. Under painful conditions, the mouse elicits below 0.3 s lasting scratching behavior (USLS) caused by chronic inflammatory stimulants. If this stimulation continues for a long time, it may become chronic neuropathic pain. IL-31 can change non-selective stimulation into itch stimulation. In contrast, PGD_2_ can change non-selective stimulation into pain stimulation transmitted by the primary nerves of C-fibers and by second-order nerves and spinothalamic tract neurons in the spinal cord. This suggests that IL-31 and PGD_2_ regulate the perception of sense (pain or itch) through their mutual functional antagonism.

## Data Availability

Not applicable.

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
