# Peer review of "Analyzing the Antinociceptive Effect of Interleukin-31 in Mice"

_ijms, 2023, doi:10.3390/ijms241411563_

Round 1

Reviewer 1 Report

This paper investigates the antinociceptive role of interleukin-31 in mice. While the results presented in the manuscript are interesting, there are several key errors in the manuscript and experimental design that lessen the potential impact of the study.

To improve the impact of the findings, the following should be taken into consideration:

1.     The Introduction section needs improvement. For example, the following extra text “Many endogenous pruritogens, such as amines, proteases, growth factors, neuropeptides, and cytokines, are locally pruritogenic when injected into the skin [6-10]. However, except for IL-31 or morphine, no other drug treatment [11]” does not help to set the aim of the research.

Some of the information provided is better suited for the Discussion section (paragraph 3), and the writing needs to be clearer. A thorough review by an English-speaking native is highly recommended.

2.     A better description of the methods is necessary, including an experimental design describing the mice groups used. It is unclear whether the data follow a Gaussian distribution.

3.     To improve Figure 1a, a 3-way ANOVA should be used, as there are three variables. For Figure 1b, a 2-way ANOVA is appropriate. To make the figures easier to understand, changing the colors to match Figure 1a is suggested.

4.     The effect of TNBC alone on LLS counts should be described. What is the effect of TNBC alone on LLS counts?

5.     In Figure 2a, the latency for the IL-31+saline group is around 10 seconds, while in Figure 1a, the group IL-31 + AEM has a latency of around 30 seconds. This discrepancy needs to be explained.

6.     To improve the analysis of Figures 2a and 2b, a 3-way ANOVA should be used in 2a and a 2-way ANOVA in 2b.

7.     The latency discrepancy for the AEM group in Figure 3a and Figure 1a needs to be addressed.

8.     To improve Figure 5a, a 3-way ANOVA should be used.

9.     Different labeling of traces in all figures could help identify which group corresponds.

10.  Reference section does not have the same format.

Minor point

Section 4.8 in Methods should be “La Jolla” instead “La Lolla”.

Overall, addressing these points will improve the clarity and impact of the study's findings.

Author Response

  1. The Introduction section needs improvement. For example, the following extra text “Many endogenous pruritogens, such as amines, proteases, growth factors, neuropeptides, and cytokines, are locally pruritogenic when injected into the skin [6-10]. However, except for IL-31 or morphine, no other drug treatment [11]” does not help to set the aim of the research.

Respons 1. Thank you for your thorough review of our manuscript. We apologize for this error; we have deleted this sentence from the revised text (lines 35-38).

As we wanted to show that IL-31 is totally different from other pruritogens, these references (6-11) moved to Discussion section. Page 15, line 529, as follows, …mine, serotonin)-induced scratching [21, 35-37].

Some of the information provided is better suited for the Discussion section (paragraph 3), and the writing needs to be clearer. A thorough review by an English-speaking native is highly recommended.

Thank you very much for your helpful suggestions for our manuscript. Following your advice, we have clarified the vague points and improved the language used in the manuscript.

  1. A better description of the methods is necessary, including an experimental design describing the mice groups used. It is unclear whether the data follow a Gaussian distribution.

Respons 2.  According to the reviewer’s suggestion, we have added a sentence in the Materials and Methods section as follows:

Page 19, lines 730 - 731. Therefore, we used C57BL/6 mice as the target group in the experiment using the IL-31RAKI mice.

  1. To improve Figure 1a, a 3-way ANOVA should be used, as there are three variables. For Figure 1b, a 2-way ANOVA is appropriate. To make the figures easier to understand, changing the colors to match Figure 1a is suggested.

Thank you for your constructive comments. Appropriate ANOVAs were performed on all data, including Figures mentioned by the reviewer, and these results were added to each Figure legends in the revised manuscript. And we have changed the figure as recommended by the reviewer.

We have added a sentence in the Figure 1 legend as follows: Page 4, lines 169 -173. Three-way ANOVA: group × treatment × time: F (5, 120) = 2.760, p = 0.0214, (a); group × treatment × time: F (23, 480) = 1.230, p = 0.2124, (c). Two-way ANOVA: group × treatment: F (1, 20) = 4.303, p = 0.0512, group: F (1, 20) = 18.68, p = 0.0003, treatment: F (1, 20) = 6.296, p = 0.0208, (b); group × treatment: F (1, 20) = 13.95, p = 0.0013, group: F (1, 20) = 14.16, p = 0.0012, treatment: F (1, 20) = 27.89, p < 0.0001, (d).

We have added a sentence in the Materials and Methods section 4.8. Statistical Analysis as follows: Page 21, lines 833 - 842. All data were statistically analyzed using GraphPad InStat and GraphPad Prism (GraphPad Software Inc., La Jolla, CA, USA). The statistical significance of differences was determined by the Student's t-test for two-group comparisons. Experimental values are expressed as means and standard errors (S.E.). Data on time-course changes in scratching counts or the percentage of mRNA expression were analyzed using one-way ANOVA followed by the Student–Newman–Keuls test. Then, other data were analyzed using one-way, two-way or three-way ANOVA followed by Bonferroni or Tukey tests. In some cases, when a main effect was significant without interaction effect, we did an exploratory and limited pairwise post-hoc comparison consistent with our a priori hypothesis. P-values less than 0.05 were considered statistically significant.

  1. The effect of TNBC alone on LLS counts should be described. What is the effect of TNBC alone on LLS counts?

Response 4. Thank you very much for pointing out this information gap. According to your comment, we have added the following sentence:

Page 3, lines 138 - 140. In the vehicle (PBS, 10 mL/kg, intraperitoneal) + AEM-treated group, a few LLS counts were observed (Fugure 1c, blue line). And these LLS counts decreased by TNCB which was not significant compared to that of the AEM-treated group (Figure 1c, yellow line).

Page 15, lins 548 - 549. These data suggest that the sensation of pain and itch may be regulated by through their functional antagonism.

  1. In Figure 2a, the latency for the IL-31+saline group is around 10 seconds, while in Figure 1a, the group IL-31 + AEM has a latency of around 30 seconds. This discrepancy needs to be explained.

Response 5. Because we tested the latency using the conventional hot-plate test (51 °C), in Figure 2 and morphine is a strong analgesic, its value differed from that in Figure 1.

Page 4, lines 175 - 178. We have changed the original sentence as follows: In the conventional hot-plate test (51 °C), morphine (3 mg/kg, subcutaneously) significantly increased the latency (Figure 2a, green line) and total antinociception index after 15–120 min administration (AUC 15120 min) (Figure 2b, green column) compared to those of the saline-treated group (Figure. 2b, blue column).

  1. To improve the analysis of Figures 2a and 2b, a 3-way ANOVA should be used in 2a and a 2-way ANOVA in 2b.

Thank you for your suggestion. Appropriate ANOVAs were performed on all data, including Figures mentioned by the reviewer, and these results were added to each Figure legends in the revised manuscript.

We have added a sentence in the Figure 2 legend as follows: Page 5, lines 201 -203. Three-way ANOVA: group × treatment × time: F (7, 168) = 2.360, p = 0.0252, (a). Two-way ANOVA: group × treatment: F (1, 20) = 12.11, p = 0.0024, group: F (1, 20) = 34.12, p < 0.0001, treatment: F (1, 20) = 5.28, p = 0.0325, (b).

  1. The latency discrepancy for the AEM group in Figure 3a and Figure 1a needs to be addressed.

Response 7: As 1% TNCB was used in the Results section 2.3, the latency rose. We have described the reason for this in the Discussion section.

Page 5, lines 208 - 210. In the TNCB (1%) applied hot-plate (45 °C) test, the latency significantly decreased from 30 to 120 min after TNCB application (Figure 3a, blue line).

Page 15, lines 566 - 568. In the present study, loxoprofen, a PGs inhibitor, significantly increased the latency and total antinociceptive activity in the TNCB (1%)-applied hot plate (45 °C) test. The application of 3% TNCB showed approximately the same result.

  1. To improve Figure 5a, a 3-way ANOVA should be used.

Thank you for your suggestion. Appropriate ANOVAs were performed on all data, including Figures mentioned by the reviewer, and these results were added to each Figure legends in the revised manuscript.

We have added a sentence in the Figure 3 legend as follows: Page 6, lines 245 -245. Two-way ANOVA: time × group: F (24, 175) = 5.382, p < 0.0001, time: F (6, 175) = 34.22, p < 0.0001, group: F (4, 175) = 37.29, p < 0.0001, (a). One-way ANOVA: F (4, 25) = 111, p < 0.0001, (b); F (3, 20) = 10.65, p = 0.0002, (c); F (3, 20) = 10.87, p = 0.0002, (d).

We have added a sentence in the Figure 4 legend as follows: Page 7, lines 271 -278. Two-way ANOVA: time × group: F (10, 90) = 14.22, p < 0.0001, time: F (5, 90) = 74.75, p < 0.0001, group: F (2, 90) = 307.4, p < 0.0001, (a); group × treatment: F (1, 20) = 18.51, p = 0.0003, group: F (1, 20) = 22.51, p = 0.0001, treatment: F (1, 20) = 16.9, p = 0.0005, (b); group × treatment: F (1, 20) = 10.65, p = 0.0039, group: F (1, 20) = 9.817, p = 0.0052, treatment: F (1, 20) = 10.65, p = 0.0039, (c); group × treatment: F (1, 20) = 3.016, p = 0.0978, group: F (1, 20) = 3.014, p = 0.0979, treatment: F (1, 20) = 3.016, p = 0.0978, (d); group × treatment: F (1, 20) = 6.93, p = 0.0160, group: F (1, 20) = 1.273, p = 0.2725, treatment: F (1, 20) = 6.932, p = 0.0159, (e).

We have added a sentence in the Figure 5 legend as follows: Page 9, lines 341 -348. Three-way ANOVA: group × treatment × time: F (5, 120) = 0.885, p = 0.4934, group × treatment: F (1, 120) = 5.631, p = 0.0192, group: F (1, 120) = 47.782, p < 0.0001, treatment: F (1, 120) = 1.591, p = 0.2096, time: F (5, 120) = 7.123, p < 0.0001, (a); group × treatment × time: F (5, 126) = 0.824, p = 0.5347, group: F (1, 126) = 903.432, p < 0.0001, treatment: F (1, 126) = 0.315, p = 0.5759, time: F (5, 126) = 50.026, p < 0.0001, (c). Two-way ANOVA: group × treatment: F (1, 20) = 1.79, p = 0.1959, group: F (1, 20) = 3.317, p = 0.0835, treatment: F (1, 20) = 4.791, p = 0.0406, (b); group × treatment: F (1, 21) = 0.2122, p = 0.6498, group: F (1, 21) = 44.57, p < 0.0001, treatment: F (1, 21) = 0.727, p = 0.4035, (d).

We have added a sentence in the Figure 6 legend as follows: Page 10, lines 377 -383. Three-way ANOVA: group × treatment × time: F (5, 120) = 1.275, p = 0.2792, group: F (1, 120) = 17.296, p < 0.0001, treatment: F (1, 120) = 49.027, p < 0.0001, time: F (5, 120) = 73.243, p < 0.0001, (a); group × treatment × time: F (7, 160) = 2.171, p = 0.0395, (c). Two-way ANOVA: group × treatment: F (1, 20) = 0.007463, p = 0.9320, group: F (1, 20) = 28.95, p < 0.0001, treatment: F (1, 20) = 10.85, p = 0.0036, (b); group × treatment: F (1, 20) = 19.79, p = 0.0002, group: F (1, 20) = 11.82, p = 0.0026, treatment: F (1, 20) = 24.87, p < 0.0001, (d).

We have added a sentence in the Figure 7 legend as follows: Page 11, lines 410 -416. Three-way ANOVA: group × treatment × time: F (5, 120) = 2.430, p = 0.0389, (a); group × treatment × time: F (5, 96) = 1.095, p = 0.3685, group: F (1, 96) = 6.646, p = 0.0115, treatment: F (1, 96) = 14.570, p = 0.0002, time: F (5, 96) = 102.550, p < 0.0001, (c). Two-way ANOVA: group × treatment: F (1, 20) = 0.4223, p = 0.5232, group: F (1, 20) = 24.67, p < 0.0001, treatment: F (1, 20) = 0.7201, p = 0.4062, (b); group × treatment: F (1, 16) = 4.854, p = 0.0426, group: F (1, 16) = 10.21, p = 0.0056, treatment: F (1, 16) = 3.69, p = 0.0727, (d).

We have added a sentence as follows: Page 17, lines 633 -635. This result shows that the antinociceptive action of loxoprofen (NSAIDs) and acetaminophen does not develop without activation of IL-31.

We have added a sentence in the Figure 8 legend as follows: Page 13, lines 478 -485. Three-way ANOVA: group × treatment × time: F (5, 126) = 1.200, p = 0.3128, group: F (1, 126) = 381.730, p < 0.0001, treatment: F (1, 126) = 6.316, p = 0.0132, time: F (5, 126) = 44.926, p < 0.0001, (a); group × treatment × time: F (5, 120) = 0.248, p = 0.9400, group: F (1, 120) = 2.609, p = 0.1089, treatment: F (1, 120) = 21.475, p < 0.0001, time: F (5, 120) = 4.594, p = 0.0007, (c). Two-way ANOVA: group × treatment: F (1, 20) = 10.33, p = 0.0043, group: F (1, 20) = 9.935, p = 0.0050, treatment: F (1, 20) = 3.516, p = 0.0754, (b); group × treatment: F (1, 20) = 10.33, p = 0.0043, group: F (1, 20) = 9.935, p = 0.0050, treatment: F (1, 20) = 3.516, p = 0.0754, (d).

We have added a sentence in the Figure 9 legend as follows: Page 14, lines 522 -526. Two-way ANOVA: group × time: F (12, 100) = 7.899, p < 0.0001, group: F (3, 100) = 18.81, p < 0.0001, time: F (4, 100) = 12.45, p < 0.0001, (a); group × time: F (10, 90) = 15.2, p < 0.0001, group: F (2, 90) = 63.57, p < 0.0001, time: F (5, 90) = 6.516, p < 0.0001, (b); group × time: F (8, 75) = 8.204, p < 0.0001, group: F (2, 75) = 31.92, p < 0.0001, time: F (4, 75) = 9.405, p < 0.0001, (c). One-way ANOVA: F (5, 30) = 12.11, p < 0.0001, (d).

  1. Different labeling of traces in all figures could help identify which group corresponds.

Response 9: We thank for your indication. We changed all figures color and added all figures captions.

  1. Reference section does not have the same format.

Response 10. Thank you for pointing out. We have ensured all References are formatted as per requested by this journal.

Minor point

Section 4.8 in Methods should be “La Jolla” instead “La Lolla”.

Overall, addressing these points will improve the clarity and impact of the study's findings.

Response 11: We apologize for this error we have now corrected the typo.

Page 19, lines 45. “La Jolla” instead “La Lolla”

In addition, a numerical mistake was found in Figure 4e, Figure 9b, and 9c, but revised these Figures while we re-examined it this time as it was minimal, and there was not general influence.

Reviewer 2 Report

Thanks for your submission.

It is an interesting study with analyzing IL-31 for pain and itching.

# It is better to revise your manuscript structure. Please place the matirials and methods between introduction and results.

# Is there a reference to use modified hot plate rather than conventional hot plate? Please describe why you choose the modofied hot plate.

# Is there a particular reason you choose roxoprofen over other NSAIDs?

# Please describe the limitiation of your study in discussion.

Author Response

Referee 2

It is an interesting study with analyzing IL-31 for pain and itching.

We thank you for your thoughtful suggestions and insights.

# It is better to revise your manuscript structure. Please place the matirials and methods between introduction and results.

Response 1: Thank you for your suggestion. In this case, we have followed the formatting requirements of the International Journal of Molecular Sciences, which ask the Materials and Methods section be placed after the Discussion.

# Is there a reference to use modified hot plate rather than conventional hot plate? Please describe why you choose the modofied hot plate.

Response 2: Recently, we have reported on the use of the modified hot-plate test on TNCB sensitization (reference 23). The modified hot-plate test is a simple method improved by the enhanced effects of TNCB. It enables the evaluation of weak analgesic drugs, which cannot be performed using the conventional hot-plate test.

# Is there a particular reason you choose roxoprofen over other NSAIDs?

Response 3: We chose Loxoprofene because it is the NSAID most frequently used in Japan.

# Please describe the limitiation of your study in discussion.

 Response 4: Thank you very much for your advice to our manuscript. According to your suggestion, we added the sentence as follows.

We have added a sentence as follows: Page 18, lines 714 - 717. In this study, we were able to show that pain and itch ing sensations may be regulated by their functional antagonism as a phenomenon side, but the adjustment mechanism involved is still unknown and should be elucidated in future studies.

In addition, a numerical mistake was found in Figure 4e, Figure 9b, and 9c, but revised these Figures while we re-examined it this time as it was minimal, and there was not general influence.

Thank you for your kindly consideration. I look forward to hearing from you.
